# A Random Matrix Analysis of Learning with $\alpha$-Dropout

**Mohamed El Amine Seddik** [1 2]  **Romain Couillet** [2 3]  **Mohamed Tamaazousti** [1]

## Abstract

This article studies a one hidden layer neural network with generalized Dropout ($\alpha$-*Dropout*), where the dropped out features are replaced with an arbitrary value $\alpha$. Specifically, under a large dimensional data and network regime, we provide the generalization performances for this network on a binary classification problem. We notably demonstrate that a careful choice of $\alpha \neq 0$ can drastically improve the generalization performances of the classifier.

## 1. Introduction

Many practical datasets contain samples with missing features which impair the behavior of machine learning models. Improperly handling these missing values results in biased predictions. While various imputation techniques exist in the literature, such as imputation of the global mean, the simplest is *zero imputation*, by which the missing features are simply replaced by zeros. Neural networks have been notably shown to be affected when trained on zero-imputed data (Hazan et al., 2015; Śmieja et al., 2018; Yi et al., 2019).

In neural networks, zero imputation can be seen as applying a Dropout (Srivastava et al., 2014) operation to the input data features, or equivalently as applying a binary mask entry-wise to the data. The Dropout operation is commonly used as a regularization technique applied to certain hidden layers of a neural network during its training phase. However, since zero imputation is known to alter the behavior of neural networks (Yi et al., 2019), the Dropout operation must result in the same deleterious effects. Dropping features with other values than zero may thus improve the Dropout in neural networks and mitigate the effects of zero imputation (Wager et al., 2013; Srinivas & Babu, 2016).

[1]CEA List, Palaiseau, France [2]Centralesupélec, Gif-sur-Yvette, France [3]University Grenoble-Alpes, Grenoble, France. Correspondence to: MEA.Seddik <mohamedelamine.seddik@cea.fr>, R.Couillet <romain.couillet@centralesupelec.fr>, M.Tamaazousti <mohamed.tamaazousti@cea.fr>.

*Presented at the first Workshop on the Art of Learning with Missing Values (Artemiss) hosted by the $37^{th}$ International Conference on Machine Learning (ICML). Copyright 2020 by the author(s).*

To prove and quantify the benefits of a $\alpha$-Dropout approach, this article studies a one hidden layer neural network with $\alpha$-Dropout, i.e., in which the missing or dropped features are replaced by a fixed real value $\alpha$. Training only the output layer, the network (sometimes referred to as an extreme learning machine (Huang et al., 2006)) reduces to a ridge-regression classifier learnt on $\alpha$-imputed data. Specifically, under the instrumental, yet instructive, setting of a network trained on a set of $n$ data samples of $p$-dimensional features (or equivalently $p$ neurons) distributed in two classes, we retrieve the exact generalization performance when both $p$ and $n$ grow large. A major outcome of our study is the identification of the optimal value of $\alpha$ which maximizes the generalization performances of the classifier.

*Notation:* $\text{Ber}(\varepsilon)$ for Bernoulli law with parameter $\varepsilon$. $\odot$ stands for the Hadamard product with $[\boldsymbol{A} \odot \boldsymbol{B}]_{ij} = A_{ij}B_{ij}$. $\boldsymbol{u}^{\odot x}$ is the vector with entries $u_i^x$. $\text{Diag}(\boldsymbol{u})$ stands for the diagonal matrix with diagonal entries $u_i$.

## 2. Model and Problem Statement

Let the training data $\boldsymbol{d}_1, \ldots, \boldsymbol{d}_n \in \mathbb{R}^q$ be independent vectors drawn from two distinct distribution classes $\mathcal{C}_1$ and $\mathcal{C}_2$ of respective cardinality $n_1$ and $n_2$ (and we denote $n = n_1 + n_2$). We suppose the $\boldsymbol{d}_i$'s pass through a first (fixed) random neural network layer with Lipschitz activation $\sigma : \mathbb{R}^q \to \mathbb{R}^p$ in such a way that $\sigma(\boldsymbol{d}_i)$ is a *concentrated random vector* (Louart & Couillet, 2018b). This random projection is then followed by a random $\alpha$-Dropout, i.e., entries of the feature vector $\sigma(\boldsymbol{d}_i)$ are dropped uniformly at random and replaced by some fixed value $\alpha \in \mathbb{R}$. Letting $\boldsymbol{\mu} \in \mathbb{R}^p$, we further suppose for simplicity of exposition that for $\boldsymbol{d}_i \in \mathcal{C}_a$, $\mathbb{E}[\sigma(\boldsymbol{d}_i)] = (-1)^a \boldsymbol{\mu}$ and $\mathbb{E}[\sigma(\boldsymbol{d}_i)\sigma(\boldsymbol{d}_i)^{\intercal}] = \boldsymbol{I}_p + \boldsymbol{\mu}\boldsymbol{\mu}^{\intercal}$.[1] Overall, after the $\alpha$-Dropout layer, the feature vector $\tilde{\boldsymbol{x}}_i \in \mathcal{C}_a$ may thus be written

$$\tilde{\boldsymbol{x}}_i = ((-1)^a \boldsymbol{\mu} + \boldsymbol{z}_i) \odot \boldsymbol{b}_i + \alpha (\mathbf{1}_p - \boldsymbol{b}_i), \qquad (1)$$

for $a \in \{1, 2\}$, where $\boldsymbol{\mu} \in \mathbb{R}^p$, $\boldsymbol{z}_i$ is a concentrated random vector with zero mean and identity covariance, and $\boldsymbol{b}_i$ is a random binary mask vector with *i.i.d.* entries $b_{ij} \sim \text{Ber}(\varepsilon)$. That is, features are discarded with an average dropout rate

---

[1]In the same vein as (Seddik et al., 2020), this assumption could be largely relaxed but simplifies the interpretation of our results.

$\varepsilon$, as performed in the classical Dropout procedure in neural networks (Srivastava et al., 2014).

The model equation 1 thus describes a single hidden layer network with $\alpha$-Dropout (dropped features are replaced by $\alpha$) applied to a two-class mixture of concentrated random vectors of mean $(-1)^a \boldsymbol{\mu}$ for $\boldsymbol{d}_i$ in class $\mathcal{C}_a$ and isotropic covariance. As shown in (Louart & Couillet, 2018b; Seddik et al., 2020), from a random matrix perspective, the asymptotic performance of the neural network under study is strictly equivalent to that of features $\tilde{\boldsymbol{x}}_i$ modelled as in equation 1 but with $\boldsymbol{z}_i \sim \mathcal{N}(0, \boldsymbol{I}_p)$, an assumption we will make from now on.

In a matrix form, the training features $\tilde{\boldsymbol{X}} = [\tilde{\boldsymbol{x}}_1, \ldots, \tilde{\boldsymbol{x}}_n] \in \mathcal{M}_{p,n}$ can be compactly written

$$\tilde{\boldsymbol{X}} = \boldsymbol{B}_\varepsilon \odot (\boldsymbol{Z} + \boldsymbol{\mu}\boldsymbol{y}^\intercal) + \alpha \left(\boldsymbol{1}_p \boldsymbol{1}_n^\intercal - \boldsymbol{B}_\varepsilon\right), \quad (2)$$

where $\boldsymbol{Z}$ has *i.i.d.* $\mathcal{N}(0,1)$ entries, $[\boldsymbol{B}_\varepsilon]_{ij} \sim \text{Ber}(\varepsilon)$ and $\boldsymbol{y} \in \mathbb{R}^n$ stands for the vector of class labels with $y_i = -1$ for $\tilde{\boldsymbol{x}}_i \in \mathcal{C}_1$ and $y_j = 1$ for $\tilde{\boldsymbol{x}}_j \in \mathcal{C}_2$.

For reasons that will be clarified in the next section, we shall consider in the rest of the paper the standardized[2] data matrix $\boldsymbol{X} \equiv \frac{\tilde{\boldsymbol{X}} \boldsymbol{P}_n}{\sqrt{\varepsilon + \alpha^2 \varepsilon(1-\varepsilon)}}$, with $\boldsymbol{P}_n = \boldsymbol{I}_n - \frac{1}{n}\boldsymbol{1}_n\boldsymbol{1}_n^\intercal$, i.e.,

$$\boldsymbol{X} = \frac{\left(\boldsymbol{B}_\varepsilon \odot (\boldsymbol{Z} + \boldsymbol{\mu}\boldsymbol{y}^\intercal)\right)\boldsymbol{P}_n + \alpha \boldsymbol{B}_\varepsilon \boldsymbol{P}_n}{\sqrt{\varepsilon + \alpha^2 \varepsilon(1-\varepsilon)}}. \quad (3)$$

Under the features data model in equation 3, we aim in the following to study the generalization performance of a (fully connected) linear layer applied to the features $\boldsymbol{x}_i$'s which is thus equivalent to optimizing (with an $\ell_2$ regularization term)

$$\mathcal{E}(\boldsymbol{w}) = \frac{1}{n}\|\boldsymbol{y} - \boldsymbol{X}^\intercal \boldsymbol{w}\|^2 + \gamma\|\boldsymbol{w}\|^2, \quad (4)$$

the solution of which is explicitly given by, for $z \in \mathbb{C} \setminus \mathbb{R}^-$

$$\boldsymbol{w} = \frac{1}{n}\boldsymbol{Q}(\gamma)\boldsymbol{X}\boldsymbol{y}, \quad \boldsymbol{Q}(z) \equiv \left(\frac{1}{n}\boldsymbol{X}\boldsymbol{X}^\intercal + z\boldsymbol{I}_p\right)^{-1}. \quad (5)$$

The associated (hard) decision function for a new datum feature vector $\boldsymbol{x} \in \mathcal{C}_a$, for $a \in \{1, 2\}$, then reads

$$g(\boldsymbol{x}) \equiv \boldsymbol{x}^\intercal \boldsymbol{w} = \frac{1}{n}\boldsymbol{x}^\intercal \boldsymbol{Q}(\gamma)\boldsymbol{X}\boldsymbol{y} \underset{\mathcal{C}_2}{\overset{\mathcal{C}_1}{\lessgtr}} 0. \quad (6)$$

The model in equation 3 coupled with the ridge loss function in equation 4 is that of an extreme learning machine trained with $\alpha$-Dropout through the random matrix $\boldsymbol{B}_\varepsilon$.

---

[2]Centring by the empirical mean and dividing by the standard deviation $\sqrt{\varepsilon + \alpha^2 \varepsilon(1-\varepsilon)}$ as in batch normalization layers (Ioffe & Szegedy, 2015).

In mathematical terms, studying the generalization performance under a large dimensional network regime consists in studying the statistical behavior of the *resolvent matrix* $\boldsymbol{Q}(z)$ defined in equation 5. The main technicality precisely arises from the unconventional presence of the matrix $\boldsymbol{B}_\varepsilon$ in the model.

Elaborating on recent tools from random matrix theory, the next section derives a *deterministic equivalent* (Hachem et al., 2007) of $\boldsymbol{Q}(z)$ which is the basic technical ingredient for the further analysis, as a function of $\alpha$ and $\varepsilon$, of the network generalization performance.

## 3. Main Results

This section establishes the asymptotic performance and the practical relevance of the $\alpha$-Dropout neural network under study. Technical preliminaries on the statistical behavior of $\boldsymbol{Q}(z)$ are first established before delving into the core analysis and main practical results of the article (in Section 3.2).

### 3.1. Deterministic Equivalent and Limiting Spectrum

Our main technical result provides a *deterministic equivalent* $\bar{\boldsymbol{Q}}(z)$ for the resolvent $\boldsymbol{Q}(z)$, that is a deterministic matrix such that, for all $\boldsymbol{A} \in \mathbb{R}^{p \times p}$ and $\boldsymbol{a}, \boldsymbol{b} \in \mathbb{R}^p$ of bounded (spectral and Euclidean, respectively) norms, with probability one,

$$\frac{1}{p}\text{Tr}\,\boldsymbol{A}\left(\boldsymbol{Q}(z) - \bar{\boldsymbol{Q}}(z)\right) \to 0, \quad \boldsymbol{a}^\intercal\left(\boldsymbol{Q}(z) - \bar{\boldsymbol{Q}}(z)\right)\boldsymbol{b} \to 0.$$

We will denote in short $\boldsymbol{Q}(z) \leftrightarrow \bar{\boldsymbol{Q}}(z)$. A large dimensional assumption will provide the existence of $\bar{\boldsymbol{Q}}(z)$ according to the following dimensional growth conditions:

**Assumption 3.1** (Growth rate). *As $n \to \infty$,*

1. *$q/n \to r \in (0, \infty)$ and $p/n \to c \in (0, \infty)$;*

2. *For $a \in \{1, 2\}$, $\frac{n_a}{n} \to c_a \in (0, 1)$;*

3. *$\|\boldsymbol{\mu}\| = \mathcal{O}(1)$.*

Under the model equation 3 and from (Hachem et al., 2007), we have:

**Proposition 3.2.** *Under Assumption 3.1,*

$$\boldsymbol{Q}(z) \leftrightarrow \bar{\boldsymbol{Q}}(z) \equiv \mathcal{D}_z - \frac{\frac{\varepsilon}{1+\alpha^2(1-\varepsilon)}\mathcal{D}_z\boldsymbol{\mu}\boldsymbol{\mu}^\intercal\mathcal{D}_z}{1 + cq(z) + \frac{\varepsilon}{1+\alpha^2(1-\varepsilon)}\boldsymbol{\mu}^\intercal\mathcal{D}_z\boldsymbol{\mu}},$$

*where $\mathcal{D}_z \equiv q(z)\,\text{Diag}\left\{\frac{1+cq(z)}{1+cq(z)+\frac{(1-\varepsilon)q(z)}{1+\alpha^2(1-\varepsilon)}\mu_i^2}\right\}_{i=1}^p$, and $q(z)$ is given by*

$$q(z) \equiv \frac{c - z - 1 + \sqrt{(c-z-1)^2 + 4zc}}{2zc}.$$

Proposition 3.2 shows that the deterministic equivalent $\bar{Q}(z)$ involves two terms: a diagonal matrix $\mathcal{D}_z$ (describing the noise part of the data model) and an informative scaled rank-1 matrix $\mathcal{D}_z \boldsymbol{\mu}\boldsymbol{\mu}^\mathsf{T}\mathcal{D}_z$. We see through the expression of $\bar{Q}(z)$ that the informative term is linked to the noise term (through $\mathcal{D}_z$) if $\varepsilon \neq 1$, and for small values of $\varepsilon$ or equivalently large values of $\alpha$ the "energy" of the informative term is transferred to the noise term which will result in a poor classification accuracy on the train set, still we will subsequently see that for a fixed value of $\varepsilon$, there exists a value of $\alpha$ which will provide optimal classification rates on the test set. We will next use the property that $\boldsymbol{a}^\mathsf{T}\boldsymbol{Q}(z)\boldsymbol{b} \simeq \boldsymbol{a}^\mathsf{T}\bar{\boldsymbol{Q}}(z)\boldsymbol{b}$ for all large $n, p$ and deterministic bounded vectors $\boldsymbol{a}, \boldsymbol{b}$, to exploit $\bar{\boldsymbol{Q}}(z)$ as a proxy for the performance analysis (which is precisely related to a bilinear form on $\boldsymbol{Q}(z)$) of the $\alpha$-Dropout neural network.

Further, let us introduce the following quantities which will be used subsequently. First, we have under Assumption 3.1, the statistics of the feature vector $\boldsymbol{x}_i$, for $\boldsymbol{x}_i \in \mathcal{C}_a$, are:

$$\boldsymbol{m}_a \equiv \mathbb{E}\left[\boldsymbol{x}_i\right] = (-1)^a \sqrt{\frac{\varepsilon}{1 + \alpha^2(1 - \varepsilon)}}\,\boldsymbol{\mu},$$

$$\boldsymbol{C}_\varepsilon \equiv \mathbb{E}\left[\boldsymbol{x}_i \boldsymbol{x}_i^\mathsf{T}\right] = \boldsymbol{I}_p + \frac{\varepsilon}{1 + \alpha^2(1 - \varepsilon)}\boldsymbol{\mu}\boldsymbol{\mu}^\mathsf{T}$$
$$+ \frac{1 - \varepsilon}{1 + \alpha^2(1 - \varepsilon)}\left(\mathrm{Diag}(\boldsymbol{\mu}^{\odot 2} + 2\alpha\boldsymbol{\mu}) - \frac{\alpha^2}{p}\boldsymbol{1}_p\boldsymbol{1}_p^\mathsf{T}\right).$$

We will also need the quantity $\delta(z) \equiv \frac{1}{n}\,\mathrm{Tr}\left(\boldsymbol{C}_\varepsilon\bar{\boldsymbol{Q}}(z)\right)$.

### 3.2. Generalization Performance of $\alpha$-Dropout

The generalization performance of the classifier relates to misclassification errors

$$P\left(g(\boldsymbol{x}) > 0 \,|\, \boldsymbol{x} \in \mathcal{C}_1\right), \quad P\left(g(\boldsymbol{x}) < 0 \,|\, \boldsymbol{x} \in \mathcal{C}_2\right)$$

where $g(\cdot)$ is the decision function previously defined in equation 6.

Since the Dropout is deactivated at inference time, the statistics of $\boldsymbol{x}$ correspond to the setting where $\varepsilon = 1$, and thus

$$\mathbb{E}[\boldsymbol{x}] = (-1)^a \boldsymbol{\mu}, \quad \boldsymbol{C}_1 = \mathbb{E}[\boldsymbol{x}\boldsymbol{x}^\mathsf{T}] = \boldsymbol{I}_p + \boldsymbol{\mu}\boldsymbol{\mu}^\mathsf{T}.$$

Further, define the following quantities which shall be used subsequently

$$\eta(\boldsymbol{A}) \equiv \frac{(1 + \delta(\gamma))\frac{1}{n}\,\mathrm{Tr}\left(\boldsymbol{C}_\varepsilon\bar{\boldsymbol{Q}}(\gamma)\boldsymbol{A}\bar{\boldsymbol{Q}}(\gamma)\right)}{(1 + \delta(\gamma))^2 - \frac{1}{n}\,\mathrm{Tr}\left(\boldsymbol{C}_\varepsilon\bar{\boldsymbol{Q}}(\gamma)\boldsymbol{C}_\varepsilon\bar{\boldsymbol{Q}}(\gamma)\right)},$$

$$\boldsymbol{\Delta}(\boldsymbol{A}) \equiv \bar{\boldsymbol{Q}}(\gamma)\left(\boldsymbol{A} + \frac{\eta(\boldsymbol{A})}{1 + \delta(\gamma)}\boldsymbol{C}_\varepsilon\right)\bar{\boldsymbol{Q}}(\gamma).$$

By Lyapunov's central limit theorem (Billingsley, 2008), the decision function has the following Gaussian approximation as $n \to \infty$.

**Theorem 3.3** (Gaussian Approximation of $g(\boldsymbol{x})$). *Under Assumption 3.1, for $\boldsymbol{x} \in \mathcal{C}_a$ with $a \in \{1, 2\}$,*

$$\nu^{-\frac{1}{2}}\left(g(\boldsymbol{x}) - m_a\right) \xrightarrow{\mathcal{D}} \mathcal{N}(0, 1)$$

*where*

$$m_a \equiv (-1)^a \sqrt{\frac{\varepsilon}{1 + \alpha^2(1 - \varepsilon)}}\,\frac{\boldsymbol{\mu}^\mathsf{T}\bar{\boldsymbol{Q}}(\gamma)\boldsymbol{\mu}}{1 + \delta(\gamma)}$$

$$\nu \equiv \frac{1}{(1 + \delta(\gamma))^2}\left(\eta(\boldsymbol{C}_1) + \frac{\varepsilon}{1 + \alpha^2(1 - \varepsilon)}\right.$$
$$\left.\times\left[\boldsymbol{\mu}^\mathsf{T}\left(\boldsymbol{\Delta}(\boldsymbol{C}_1) - \bar{\boldsymbol{Q}}(\gamma)\right)\boldsymbol{\mu} - \frac{2\,\eta(\boldsymbol{C}_1)\boldsymbol{\mu}^\mathsf{T}\bar{\boldsymbol{Q}}(\gamma)\boldsymbol{\mu}}{1 + \delta(\gamma)}\right]\right).$$

In a nutshell, Theorem 3.3 states that the one hidden layer network classifier with $\alpha$-Dropout is asymptotically equivalent to the thresholding of two monovariate Gaussian random variables, the means and variances of which depend on $\boldsymbol{\mu}, \boldsymbol{C}_\varepsilon$ and the parameters $\alpha$ and $\varepsilon$. As such, we have the corresponding (asymptotic) classification errors:

**Corollary 3.4** (Generalization Performance of $\alpha$-Dropout). *Under the setting of Theorem 3.3, for $a \in \{1, 2\}$, with probability one*

$$P\left((-1)^a\, g(\boldsymbol{x}) < 0 \,|\, \boldsymbol{x} \in \mathcal{C}_a\right) - Q\left(\frac{m_a}{\sqrt{\nu}}\right) \to 0$$

*with $Q(x) = \frac{1}{\sqrt{2\pi}}\int_x^\infty e^{-u^2/2}du$ the Gaussian tail function.*

Corollary 3.4 can therefore be exploited to find the optimal value of $\alpha^*$ which minimizes the test misclassification error, since $Q'(x) < 0$ the optimal value $\alpha^*$ satisfies the equation $\frac{1}{m_a}\frac{\partial m_a}{\partial \alpha} = \frac{1}{\sqrt{\nu}}\frac{\partial\sqrt{\nu}}{\partial \alpha}$ which can be solved numerically.

### 3.3. Training Performance of $\alpha$-Dropout

It is instructive to compare the generalization versus training performances of the network classifier with $\alpha$-Dropout. Following similar arguments as in (Louart & Couillet, 2018b), the central limit argument of the previous section also holds for $g(\boldsymbol{x})$ with $\boldsymbol{x} \in \mathcal{C}_a$ taken from the training set $\boldsymbol{X}$.

**Theorem 3.5** (Training performance of $\alpha$-Dropout). *Under Assumption 3.1, for $\boldsymbol{x} \in \mathcal{C}_a$ with $a \in \{1, 2\}$ a column of $\boldsymbol{X}$, with probability one,*

$$P\left((-1)^a\, g(\boldsymbol{x}) < 0 \,|\, \boldsymbol{x} \in \mathcal{C}_a\right) - Q\left(\frac{\bar{m}_a}{\sqrt{\bar{\nu} - \bar{m}_a^2}}\right) \to 0$$

*where*

$$\bar{m}_a \equiv \frac{\delta(\gamma)}{1 + \delta(\gamma)} + \frac{(-1)^a\varepsilon}{1 + \alpha^2(1 - \varepsilon)}\frac{\boldsymbol{\mu}^\mathsf{T}\bar{\boldsymbol{Q}}(\gamma)\boldsymbol{\mu}}{(1 + \delta(\gamma))^2}$$

$$\bar{\nu} \equiv \left(\frac{\delta(\gamma)}{1 + \delta(\gamma)}\right)^2 + \frac{\eta(\boldsymbol{C}_\varepsilon)}{(1 + \delta(\gamma))^4} + \frac{\varepsilon}{1 + \alpha^2(1 - \varepsilon)}$$
$$\times\boldsymbol{\mu}^\mathsf{T}\left(\frac{\delta(\gamma)\bar{\boldsymbol{Q}}(\gamma)}{(1 + \delta(\gamma))^3} + \frac{\boldsymbol{\Delta}(\boldsymbol{C}_\varepsilon)}{(1 + \delta(\gamma))^4} - \frac{2\,\eta(\boldsymbol{C}_\varepsilon)\bar{\boldsymbol{Q}}}{(1 + \delta(\gamma))^5}\right)\boldsymbol{\mu}.$$

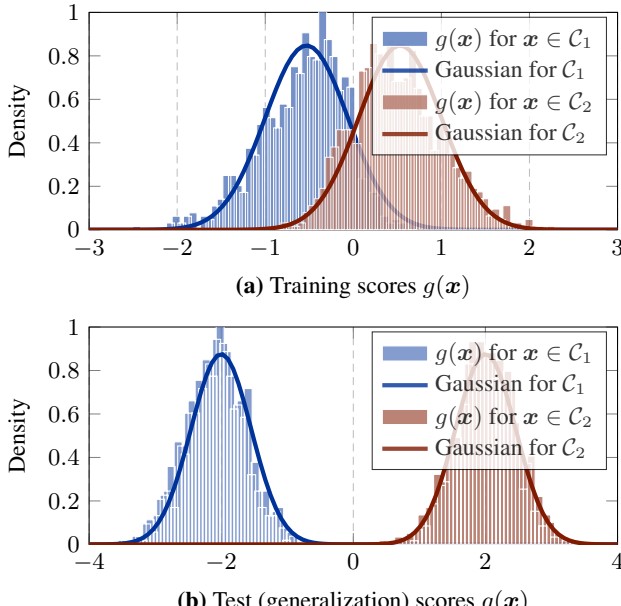

**(a)** Training scores $g(\boldsymbol{x})$

**(b)** Test (generalization) scores $g(\boldsymbol{x})$

Figure 1: Histogram of the decision function $g(\boldsymbol{x})$ when applied to the training data **(a)** and test data **(b)**. The curves represent the Gaussian approximations as per Theorem 3.3 and Theorem 3.5 for test and training data respectively. We used the parameters $\boldsymbol{\mu} = \frac{5 \cdot \boldsymbol{u}}{\|\boldsymbol{u}\|}$ with $\boldsymbol{u} = [10, 10, -10, -10, \boldsymbol{v}]$ where $\boldsymbol{v} \sim \mathcal{N}(\boldsymbol{0}, \boldsymbol{I}_{p-4})$, $p = 125$, $n_1 = n_2 = 1000$, $\gamma = 1 \cdot 10^{-2}$, $\varepsilon = 0.25$ and $\alpha = 2$.

## 4. Simulations

### 4.1. Gaussian Approximations of the Decision Function

We complete this article by simulations to validate our theoretical findings. Figure 1 depicts histograms showing the distribution of $g(\boldsymbol{x})$ for both **(a)** training and **(b)** test data. As we can see, these distributions are well approximated by monovariate Gaussians as per Theorem 3.3 and Theorem 3.5. Since the $\alpha$-Dropout removes features at random from the training data, the misclassification error happens to be larger on the training set compared to the test set. Notably, the difference between the training and test error arises theoretically from the term $\kappa \equiv \sqrt{\frac{\varepsilon}{1 + \alpha^2(1 - \varepsilon)}}$ as $\bar{m}_a \approx \kappa m_a$, therefore, for small values of $\varepsilon$ the training error is larger than the test error, which shows the regularization effect of the Dropout.

### 4.2. Training and Test Performances

Figure 2 depicts the theoretical **(a)** training (through Theorem 3.5) and **(b)** test (through Corollary 3.4) misclassification errors, for different values of $\varepsilon$ and in terms of $\alpha$, and their simulated counterparts. We can notice from these plots that the training error increases with $\alpha$ and is minimal for

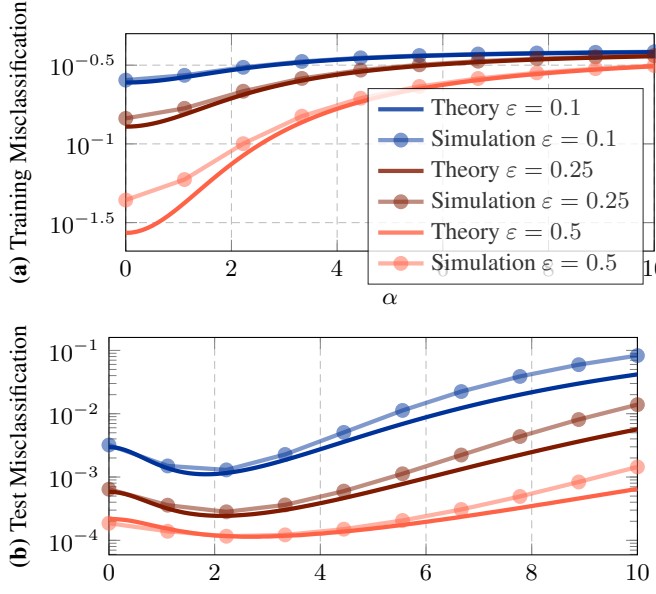

**(a)** Training Misclassification

**(b)** Test Misclassification

Figure 2: **(a)** Training and **(b)** Test misclassification errors as per Theorem 3.5 and Corollary 3.4 respectively. We used the parameters $\boldsymbol{\mu} = 4 \cdot [\frac{1}{\sqrt{4}}, \frac{1}{\sqrt{4}}, -\frac{1}{\sqrt{4}}, -\frac{1}{\sqrt{4}}, \boldsymbol{0}_{p-4}^{\mathsf{T}}]^{\mathsf{T}}$, $p = 125$, $n_1 = n_2 = 1000$ and $\gamma = 1 \cdot 10^{-2}$. Simulations are obtained through 100 Monte-Carlo runs of independent realizations of the matrix $\boldsymbol{X}$ as in equation 3.

$\alpha = 0$. In contrast, the test misclassifaction error is convex in terms of $\alpha$ and therefore the lowest generalization error corresponds to an optimal value $\alpha \neq 0$. We also remark that the optimal value of $\alpha$ increases in terms of $\varepsilon$ which is counterintuitive since we excpect and $\alpha$ near to 0 for large values of $\varepsilon$, but actually, the test misclassification error in terms of $\alpha$ gets more and more flatter as $\varepsilon$ increases.

## 5. Conclusion and Discussion

Leveraging on random matrix theory, we have analyzed the effect of the $\alpha$-Dropout layer on a one layer neural network, which allowed us to have a deeper understanding of the impact of this layer. We have notably exhibited an optimal Dropout operation (dropping our features with some $\alpha \neq 0$) in terms of the generalization error of the studied classifier. Although, our analysis was presented on a simple binary classification task, it can be straightforwardly generalized to a more realistic data model as the mixture of $k$-class model (Louart & Couillet, 2018a; Seddik et al., 2020). Under a $k$-class model it may be beneficial to consider an $\alpha_\ell$ per class $\mathcal{C}_\ell$ as the classes may be constructed with different statistics. Following the same approach one can derive the test misclassification error as per Corollary 3.4 in terms of scalar quantities involving the data statistics, and therefore exploit the formulas to find the optimal values of $\alpha_\ell$'s.

## Acknowledgements

This work is supported by the MIAI LargeDATA Chair at University Grenoble-Alpes and the GIPSA-HUAWEI Labs project Lardist.

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

## A. Proofs

**Lemma A.1** (Woodbury Identity).

$$(A + uu^\mathsf{T})^{-1} = A^{-1} - \frac{A^{-1}uu^\mathsf{T}A^{-1}}{1 + u^\mathsf{T}A^{-1}u} \tag{7}$$

### A.1. Statistics of $x_i$ in equation 3.1

*Statistics of $x_i$ in equation 3.1.* Let $x = b \odot (z + \mu) + \alpha Pb$. Denote $\mathbb{E}[Pb] = \mathbb{E}[b - \frac{1}{p}\sum_{i=1}^n b_i \mathbf{1}_p] = \mathbf{0}$, we have $\mathbb{E}[b \odot (z + \mu)] = b \odot \mu$ and $\mathbb{E}[bb^\mathsf{T}] = \varepsilon \mathbf{1}_p \mathbf{1}_p^\mathsf{T} + (1 - \varepsilon)I_p \equiv C_b$. Therefore,

$$\mathbb{E}[\alpha Pb(\alpha Pb)^\mathsf{T}] = \alpha^2 PC_bP^\mathsf{T}$$
$$\mathbb{E}[b \odot (z + \mu)(b \odot (z + \mu))^\mathsf{T}] = C_b \odot (I_p + \mu\mu^\mathsf{T})$$
$$\mathbb{E}[b \odot (z + \mu)(\alpha Pb)^T] = \alpha \operatorname{Diag}(\mu)C_bP$$

Hence,

$$\mathbb{E}[xx^\mathsf{T}] = C_b \odot (I_p + \mu\mu^\mathsf{T}) + \alpha \operatorname{Diag}(\mu)C_bP + \alpha PC_b \operatorname{Diag}(\mu) + \alpha^2 PC_bP^\mathsf{T}$$

Since $\|\mu\| = \mathcal{O}(1)$, as in Assumption 3.1, we have

$$C_b \odot (I_p + \mu\mu^\mathsf{T}) = \varepsilon I_p + (1 - \varepsilon)I_p + \varepsilon\mu\mu^\mathsf{T} + (1 - \varepsilon)\operatorname{Diag}(\mu^{\odot 2}) = I_p + \varepsilon\mu\mu^\mathsf{T} + (1 - \varepsilon)\operatorname{Diag}(\mu^{\odot 2})$$
$$\operatorname{Diag}(\mu)C_bP = (1 - \varepsilon)\operatorname{Diag}(\mu)P = (1 - \varepsilon)\operatorname{Diag}(\mu) - \frac{1 - \varepsilon}{p}\mu\mathbf{1}_p^\mathsf{T} = (1 - \varepsilon)\operatorname{Diag}(\mu) + \mathcal{O}_{\|\cdot\|}(p^{-\frac{1}{2}})$$
$$PC_bP = (1 - \varepsilon)P$$

Thus

$$\mathbb{E}[xx^\mathsf{T}] = (1 + \alpha^2(1 - \varepsilon))I_p + \varepsilon\mu\mu^\mathsf{T} - \frac{\alpha^2(1 - \varepsilon)}{p}\mathbf{1}_p\mathbf{1}_p^\mathsf{T} + (1 - \varepsilon)\operatorname{Diag}(\mu^{\odot 2}) + 2\alpha(1 - \varepsilon)\operatorname{Diag}(\mu) + \mathcal{O}_{\|\cdot\|}(p^{-\frac{1}{2}})$$

Therefore,

$$m_a \equiv \mathbb{E}[x_i] = (-1)^a \sqrt{\frac{\varepsilon}{1 + \alpha^2(1 - \varepsilon)}}\mu \tag{8}$$

$$C_\varepsilon \equiv \mathbb{E}[x_i x_i^\mathsf{T}] = I_p + \frac{\varepsilon}{1 + \alpha^2(1 - \varepsilon)}\mu\mu^\mathsf{T}\frac{1 - \varepsilon}{1 + \alpha^2(1 - \varepsilon)}\left(\operatorname{Diag}(\mu^{\odot 2}) + 2\alpha\operatorname{Diag}(\mu)\right) \tag{9}$$

$$- \frac{\alpha^2(1 - \varepsilon)}{p(1 + \alpha^2(1 - \varepsilon))}\mathbf{1}_p\mathbf{1}_p^\mathsf{T} + \mathcal{O}_{\|\cdot\|}(p^{-\frac{1}{2}}). \tag{10}$$

$\square$

### A.2. Proof of Proposition 3.2

*Proof of Proposition 3.2.* Let $A = \frac{P(B_\varepsilon \odot (\mu y^\mathsf{T}))}{\sqrt{\varepsilon + \alpha^2 \varepsilon(1 - \varepsilon)}}$, form (Hachem et al., 2007), a deterministic equivalent of $Q(z)$ is given by

$$\bar{Q}(z) \equiv \left(q^{-1}(z) + \frac{1}{1 + cq(z)}\frac{1}{n}\mathbb{E}[AA^\mathsf{T}]\right)^{-1} \quad \text{with} \quad q(z) \equiv \frac{1 + cq(z)}{1 + z(1 + cq(z))}$$

And by Assumption 3.1 ($\|\mu\| = \mathcal{O}(1)$), we have

$$\frac{1}{n}\mathbb{E}[AA^\mathsf{T}] = \frac{\varepsilon}{1 + \alpha^2(1 - \varepsilon)}\mu\mu^\mathsf{T} + \frac{1 - \varepsilon}{1 + \alpha^2(1 - \varepsilon)}\operatorname{Diag}(\mu^{\odot 2}) + \mathcal{O}_{\|\cdot\|}(n^{-\frac{1}{2}})$$
$$= a\mu\mu^\mathsf{T} + b\operatorname{Diag}(\mu^{\odot 2}) + \mathcal{O}_{\|\cdot\|}(p^{-\frac{1}{2}})$$

Therefore, by Lemma A.1, we have (we denote $r(z) = \frac{1}{1+cq(z)}$)

$$\bar{Q}(z) = \left(q^{-1}I_p + br(z)\operatorname{Diag}(\mu^{\odot 2}) + ar(z)\mu\mu^\intercal\right)^{-1}$$

$$= \underbrace{\left(q^{-1}I_p + br(z)\operatorname{Diag}(\mu^{\odot 2})\right)^{-1}}_{\mathcal{D}_z} - \frac{ar(z)\mathcal{D}_z\mu\mu^\intercal\mathcal{D}_z}{1 + ar(z)\mu^\intercal\mathcal{D}_z\mu}$$

Finally solving the second order equation $q(z) = \frac{1+cq(z)}{1+z(1+cq(z))}$ in $q(z)$ completes the proof. $\qquad\square$

### A.3. Proof of Theorem 3.3

*Proof of Theorem 3.3.* For clarity, we simply write $Q(z) = Q$ and $\delta(z) = \delta$ removing the dependence on $z$, and let the resolvent $Q_{-i}$ which is $Q$ without the $i$-th datum $x_i$ defined as

$$Q_{-i} = \left(\frac{1}{n}XX^\intercal - \frac{1}{n}x_ix_i^\intercal + zI_p\right)^{-1} \tag{11}$$

And let

$$m \equiv \sqrt{\frac{\varepsilon}{1 + \alpha^2(1-\varepsilon)}}\mu \tag{12}$$

**Estimation of $\mathbb{E}[g(x)]$** Using the identity $Qx_i = \frac{Q_{-i}x_i}{1+\frac{1}{n}x_i^\intercal Q_{-i}x_i}$, we have for $x \in \mathcal{C}_a$

$$\mathbb{E}\left[x^\intercal w\right] = \frac{1}{n}\mathbb{E}\left[x^\intercal QXy\right] = \frac{1}{n}\sum_{i=1}^n y_i\,\mathbb{E}\left[x^\intercal Qx_i\right] = \frac{1}{n}\sum_{i=1}^n y_i\,\mathbb{E}\left[\frac{x^\intercal Q_{-i}x_i}{1+\frac{1}{n}x_i^\intercal Q_{-i}x_i}\right]$$

$$= \frac{1}{n}\sum_{i=1}^n y_i\,\mathbb{E}\left[\frac{x^\intercal Q_{-i}x_i}{1+\delta}\right] + \mathcal{O}\left(\frac{1}{\sqrt{n}}\right) = (-1)^a\frac{\mu^\intercal\bar{Q}m}{1+\delta} + \mathcal{O}\left(\frac{1}{\sqrt{n}}\right)$$

**Estimation of $\mathbb{E}[g(x)^2]$**

$$\mathbb{E}\left[(x^\intercal w)^2\right] = \frac{1}{n^2}\mathbb{E}\left[y^\intercal X^\intercal Qxx^\intercal QXy\right] = \frac{1}{n^2}\mathbb{E}\left[y^\intercal X^\intercal QC_1QXy\right]$$

$$= \frac{1}{n^2}\sum_{i,j=1}^n y_iy_j\,\mathbb{E}\left[x_i^\intercal QC_1Qx_j\right] = \frac{1}{n^2}\sum_{i=1}^n y_i^2\,\mathbb{E}\left[x_i^\intercal QC_1Qx_i\right] + \frac{1}{n^2}\sum_{i\neq j}y_iy_j\,\mathbb{E}\left[x_i^\intercal QC_1Qx_j\right]$$

$$= \frac{1}{n^2}\sum_{i=1}^n y_i^2\,\mathbb{E}\left[\frac{x_i^\intercal Q_{-i}C_1Q_{-i}x_i}{(1+\delta)^2}\right] + \frac{1}{n^2}\sum_{i\neq j}y_iy_j\,\mathbb{E}\left[\frac{x_i^\intercal Q_{-i}C_1Q_{-j}x_j}{(1+\delta)^2}\right] + \mathcal{O}\left(\frac{1}{\sqrt{n}}\right)$$

$$= \frac{1}{n}\frac{\operatorname{Tr}\left(C\,\mathbb{E}\left[Q_{-i}C_1Q_{-i}\right]\right)}{(1+\delta)^2} + \frac{1}{n^2}\sum_{i\neq j}y_iy_j\,\mathbb{E}\left[\frac{x_i^\intercal Q_{-i}C_1Q_{-j}x_j}{(1+\delta)^2}\right] + \mathcal{O}\left(\frac{1}{\sqrt{n}}\right)$$

And using the identity $Q = Q_{-i} - \frac{Q_{-i}\frac{1}{n}x_ix_i^\intercal Q_{-i}}{1+\frac{1}{n}x_i^\intercal Q_{-i}x_i}$, the second term develops as

$$\frac{1}{n^2}\sum_{i\neq j}y_iy_j\,\mathbb{E}\left[\frac{x_i^\intercal Q_{-i}C_1Q_{-j}x_j}{(1+\delta)^2}\right]$$

$$= \frac{1}{n^2}\sum_{i\neq j}y_iy_j\,\mathbb{E}\left[\frac{x_i^\intercal Q_{-ij}C_1Q_{-ji}x_j}{(1+\delta)^2}\right] - \frac{1}{n^3}\sum_{i\neq j}y_iy_j\,\mathbb{E}\left[\frac{x_i^\intercal Q_{-ij}C_1Q_{-ji}x_ix_i^\intercal Q_{-ji}x_j}{(1+\delta)^3}\right]$$

$$- \frac{1}{n^3}\sum_{i\neq j}y_iy_j\,\mathbb{E}\left[\frac{x_i^\intercal Q_{-ji}x_jx_j^\intercal Q_{-ij}C_1Q_{-ji}x_j}{(1+\delta)^3}\right] + \frac{1}{n^4}\sum_{i\neq j}y_iy_j\,\mathbb{E}\left[\frac{x_i^\intercal Q_{-ij}x_jx_j^\intercal Q_{-ij}C_1Q_{-ji}x_ix_i^\intercal Q_{-ij}x_j}{(1+\delta)^4}\right] + \mathcal{O}\left(\frac{1}{\sqrt{n}}\right)$$

$$= \frac{m^\intercal\mathbb{E}[Q_{-ij}C_1Q_{-ji}]m}{(1+\delta)^2} - \frac{2\operatorname{Tr}(\mathbb{E}[CQC_1Q])}{n(1+\delta)^3}m^\intercal\bar{Q}m + \frac{1}{n^2(1+\delta)^4}(m^\intercal\bar{Q}m)^2m^\intercal\mathbb{E}[QC_1Q]m + \mathcal{O}\left(\frac{1}{\sqrt{n}}\right)$$

where the term $\mathbb{E}\left[\boldsymbol{Q}\boldsymbol{A}\boldsymbol{Q}\right]$ is handled by

$$\eta(\boldsymbol{A}) \equiv \frac{1}{n}\operatorname{Tr}\left(\boldsymbol{C}_\varepsilon \mathbb{E}\left[\boldsymbol{Q}\boldsymbol{A}\boldsymbol{Q}\right]\right) = \frac{(1+\delta)\frac{1}{n}\operatorname{Tr}\left(\boldsymbol{C}_\varepsilon \bar{\boldsymbol{Q}}\boldsymbol{A}\bar{\boldsymbol{Q}}\right)}{(1+\delta)^2 - \frac{1}{n}\operatorname{Tr}\left(\boldsymbol{C}_\varepsilon \bar{\boldsymbol{Q}}\boldsymbol{C}_\varepsilon \bar{\boldsymbol{Q}}\right)}$$

$$\boldsymbol{\Delta}(\boldsymbol{A}) \equiv \mathbb{E}\left[\boldsymbol{Q}\boldsymbol{A}\boldsymbol{Q}\right] = \bar{\boldsymbol{Q}}\boldsymbol{A}\bar{\boldsymbol{Q}} + \frac{\eta(\boldsymbol{A})}{1+\delta}\bar{\boldsymbol{Q}}\boldsymbol{C}_\varepsilon \bar{\boldsymbol{Q}}$$

Putting all together yields to

$$\mathbb{E}\left[(\boldsymbol{x}^\mathsf{T}\boldsymbol{w})^2\right] = \frac{\eta(\boldsymbol{C}_1)}{(1+\delta)^2} + \frac{\boldsymbol{m}^\mathsf{T}\boldsymbol{\Delta}(\boldsymbol{C}_1)\boldsymbol{m}}{(1+\delta)^2} - \frac{2\eta(\boldsymbol{C}_1)\boldsymbol{m}^\mathsf{T}\bar{\boldsymbol{Q}}\boldsymbol{m}}{(1+\delta)^3} + \mathcal{O}\left(\frac{1}{n^2}\right)$$

$\square$

## A.4. Proof of Theorem 3.5

*Proof of Theorem 3.5.* Using the previous notations and matrix identities, for $\boldsymbol{x}_i \in \mathcal{C}_a$ a sample from the training set $\boldsymbol{X}$, we have:

**Estimation of $\mathbb{E}[g(\boldsymbol{x}_i)]$**

$$\mathbb{E}[\boldsymbol{x}_i^\mathsf{T}\boldsymbol{w}] = \frac{1}{n}\mathbb{E}[\boldsymbol{x}_i^\mathsf{T}\boldsymbol{Q}\boldsymbol{X}\boldsymbol{y}] = \frac{1}{n}\sum_{j=1}^{n}y_j\,\mathbb{E}[\boldsymbol{x}_i^\mathsf{T}\boldsymbol{Q}\boldsymbol{x}_j]$$

$$= \frac{1}{n}\mathbb{E}\left[\frac{\boldsymbol{x}_i^\mathsf{T}\boldsymbol{Q}_{-i}\boldsymbol{x}_i}{1+\frac{1}{n}\boldsymbol{x}_i^\mathsf{T}\boldsymbol{Q}_{-i}\boldsymbol{x}_i}\right] + \frac{1}{n}\sum_{j\neq i}y_j\,\mathbb{E}\left[\frac{\boldsymbol{x}_i^\mathsf{T}\boldsymbol{Q}_{-ji}\boldsymbol{x}_j}{(1+\delta)^2}\right] + \mathcal{O}\left(\frac{1}{\sqrt{n}}\right)$$

$$= \frac{\delta}{1+\delta} + (-1)^a\frac{\boldsymbol{m}^\mathsf{T}\bar{\boldsymbol{Q}}\boldsymbol{m}}{(1+\delta)^2} + \mathcal{O}\left(\frac{1}{\sqrt{n}}\right)$$

**Estimation of $\mathbb{E}[g(\boldsymbol{x}_i)^2]$**

$$\mathbb{E}\left[(\boldsymbol{x}_i^\mathsf{T}\boldsymbol{w})^2\right] = \frac{1}{n^2}\mathbb{E}\left[\boldsymbol{y}^\mathsf{T}\boldsymbol{X}^\mathsf{T}\boldsymbol{Q}\boldsymbol{x}_i\boldsymbol{x}_i^\mathsf{T}\boldsymbol{Q}\boldsymbol{X}\boldsymbol{y}\right] = \frac{1}{n^2}\mathbb{E}\left[\frac{\boldsymbol{y}^\mathsf{T}\boldsymbol{X}^\mathsf{T}\boldsymbol{Q}_{-i}\boldsymbol{x}_i\boldsymbol{x}_i^\mathsf{T}\boldsymbol{Q}_{-i}\boldsymbol{X}\boldsymbol{y}}{(1+\delta)^2}\right] + \mathcal{O}\left(\frac{1}{\sqrt{n}}\right)$$

$$= \frac{1}{n^2}\sum_{j,k=1}^{n}y_j y_k\,\mathbb{E}\left[\frac{\boldsymbol{x}_j^\mathsf{T}\boldsymbol{Q}_{-i}\boldsymbol{x}_i\boldsymbol{x}_i^\mathsf{T}\boldsymbol{Q}_{-i}\boldsymbol{x}_k}{(1+\delta)^2}\right] + \mathcal{O}\left(\frac{1}{\sqrt{n}}\right)$$

$$= \frac{1}{n^2}\sum_{j=1}^{n}y_j^2\,\mathbb{E}\left[\frac{\boldsymbol{x}_j^\mathsf{T}\boldsymbol{Q}_{-i}\boldsymbol{x}_i\boldsymbol{x}_i^\mathsf{T}\boldsymbol{Q}_{-i}\boldsymbol{x}_j}{(1+\delta)^2}\right] + \frac{1}{n^2}\sum_{j\neq k}y_j y_k\,\mathbb{E}\left[\frac{\boldsymbol{x}_j^\mathsf{T}\boldsymbol{Q}_{-i}\boldsymbol{x}_i\boldsymbol{x}_i^\mathsf{T}\boldsymbol{Q}_{-i}\boldsymbol{x}_k}{(1+\delta)^2}\right] + \mathcal{O}\left(\frac{1}{\sqrt{n}}\right)$$

$$= \frac{1}{n^2}\mathbb{E}\left[\frac{(\boldsymbol{x}_i^\mathsf{T}\boldsymbol{Q}_{-i}\boldsymbol{x}_i)^2}{(1+\delta)^2}\right] + \frac{1}{n^2}\sum_{j\neq i}y_j^2\,\mathbb{E}\left[\frac{\boldsymbol{x}_j^\mathsf{T}\boldsymbol{Q}_{-i}\boldsymbol{x}_i\boldsymbol{x}_i^\mathsf{T}\boldsymbol{Q}_{-i}\boldsymbol{x}_j}{(1+\delta)^2}\right] + \frac{1}{n^2}\sum_{j\neq i}y_j y_i\,\mathbb{E}\left[\frac{\boldsymbol{x}_j^\mathsf{T}\boldsymbol{Q}_{-i}\boldsymbol{x}_i\boldsymbol{x}_i^\mathsf{T}\boldsymbol{Q}_{-i}\boldsymbol{x}_i}{(1+\delta)^2}\right]$$

$$+ \frac{1}{n^2}\sum_{i\neq j\neq k}y_j y_k\,\mathbb{E}\left[\frac{\boldsymbol{x}_j^\mathsf{T}\boldsymbol{Q}_{-i}\boldsymbol{x}_i\boldsymbol{x}_i^\mathsf{T}\boldsymbol{Q}_{-i}\boldsymbol{x}_k}{(1+\delta)^2}\right] + \mathcal{O}\left(\frac{1}{\sqrt{n}}\right)$$

$$= \left(\frac{\delta}{1+\delta}\right)^2 + \frac{1}{n^2}\sum_{j\neq i}\mathbb{E}\left[\frac{\boldsymbol{x}_j^\mathsf{T}\boldsymbol{Q}_{-ij}\boldsymbol{x}_i\boldsymbol{x}_i^\mathsf{T}\boldsymbol{Q}_{-ij}\boldsymbol{x}_j}{(1+\delta)^4}\right] + \frac{1}{n^2}\sum_{j\neq i}y_j y_i\,\mathbb{E}\left[\frac{\boldsymbol{x}_j^\mathsf{T}\boldsymbol{Q}_{-ij}\boldsymbol{x}_i\boldsymbol{x}_i^\mathsf{T}\boldsymbol{Q}_{-i}\boldsymbol{x}_i}{(1+\delta)^3}\right]$$

$$+ \frac{1}{n^2}\sum_{i\neq j\neq k}y_j y_k\,\mathbb{E}\left[\frac{\boldsymbol{x}_j^\mathsf{T}\boldsymbol{Q}_{-ij}\boldsymbol{x}_i\boldsymbol{x}_i^\mathsf{T}\boldsymbol{Q}_{-ik}\boldsymbol{x}_k}{(1+\delta)^4}\right] + \mathcal{O}\left(\frac{1}{\sqrt{n}}\right)$$

$$= \left(\frac{\delta}{1+\delta}\right)^2 + \frac{\frac{1}{n}\operatorname{Tr}\left(\boldsymbol{C}_\varepsilon \mathbb{E}[\boldsymbol{Q}_{-ij}\boldsymbol{C}_\varepsilon \boldsymbol{Q}_{-ij}]\right)}{(1+\delta)^4} + \frac{\delta\boldsymbol{m}^\mathsf{T}\bar{\boldsymbol{Q}}\boldsymbol{m}}{(1+\delta)^3}$$

$$+ \frac{1}{n^2}\sum_{i\neq j\neq k}y_j y_k\,\frac{\mathbb{E}\left[\boldsymbol{x}_j^\mathsf{T}\boldsymbol{Q}_{-ij}\boldsymbol{C}_\varepsilon \boldsymbol{Q}_{-ik}\boldsymbol{x}_k\right]}{(1+\delta)^4} + \mathcal{O}\left(\frac{1}{\sqrt{n}}\right)$$

where we have previously estimated the term $\frac{1}{n^2} \sum_{i \neq j \neq k} y_j y_k \, \mathbb{E}[\boldsymbol{x}_j^\mathsf{T} \boldsymbol{Q}_{-ij} \boldsymbol{C}_\varepsilon \boldsymbol{Q}_{-ik} \boldsymbol{x}_k]$ as

$$\frac{1}{n^2} \sum_{i \neq j \neq k} y_j y_k \, \mathbb{E}[\boldsymbol{x}_j^\mathsf{T} \boldsymbol{Q}_{-ij} \boldsymbol{C}_\varepsilon \boldsymbol{Q}_{-ik} \boldsymbol{x}_k] = \boldsymbol{m}^\mathsf{T} \, \mathbb{E}[\boldsymbol{Q}_{-ijk} \boldsymbol{C}_\varepsilon \boldsymbol{Q}_{-ijk}] \boldsymbol{m} - \frac{2 \operatorname{Tr}(\mathbb{E}[\boldsymbol{C}_\varepsilon \boldsymbol{Q} \boldsymbol{C}_\varepsilon \boldsymbol{Q}])}{n(1 + \delta)} \boldsymbol{m}^\mathsf{T} \bar{\boldsymbol{Q}} \boldsymbol{m} + \mathcal{O}\left( \frac{1}{\sqrt{n}} \right)$$

Hence, putting all together we get

$$\mathbb{E}[g(\boldsymbol{x}_i)^2] = \left( \frac{\delta}{1 + \delta} \right)^2 + \frac{\eta(\boldsymbol{C}_\varepsilon)}{(1 + \delta)^4} + \frac{\delta \boldsymbol{m}^\mathsf{T} \bar{\boldsymbol{Q}} \boldsymbol{m}}{(1 + \delta)^3} + \frac{\boldsymbol{m}^\mathsf{T} \boldsymbol{\Delta}(\boldsymbol{C}_\varepsilon) \boldsymbol{m}}{(1 + \delta)^4} - \frac{2\eta \boldsymbol{m}^\mathsf{T} \bar{\boldsymbol{Q}} \boldsymbol{m}}{(1 + \delta)^5} + \mathcal{O}\left( \frac{1}{\sqrt{n}} \right)$$

$$= \left( \frac{\delta}{1 + \delta} \right)^2 + \frac{\eta(\boldsymbol{C}_\varepsilon)}{(1 + \delta)^4} + \boldsymbol{m}^\mathsf{T} \left( \frac{\delta \bar{\boldsymbol{Q}}}{(1 + \delta)^3} + \frac{\boldsymbol{\Delta}(\boldsymbol{C}_\varepsilon)}{(1 + \delta)^4} - \frac{2\eta(\boldsymbol{C}_\varepsilon) \bar{\boldsymbol{Q}}}{(1 + \delta)^5} \right) \boldsymbol{m} + \mathcal{O}\left( \frac{1}{\sqrt{n}} \right)$$

and the CLT is obtained with similar arguments than (Louart & Couillet, 2018b). $\qquad\square$