# OpenReview forum: "A Random Matrix Analysis of Learning with α-Dropout"
_ICML.cc/2020/Workshop/Artemiss — ICML Artemiss 2020_

### Official Review · AnonReviewer1 · 2020-06-23
**Theoretical analysis a a simplified dropout model**

**Confidence:** 3
**Rating:** 6

**Review:**

The authors consider the following setup: binary classification with a 2-layer fully connected net whose first layer is random, with an alpha-dropout layer in the middle.

Under strong assumptions on the data generating process (the authors assume that the output of the first layer is a Gaussian mixture), the authors study the decision boudary and generalisation performance of learning with alpha dropout.

Some comments:
- The paper is nicely written
- Corollary 3.4 indicates the exitence of an optimal alpha and suggest a way to find. The practical consequences of this should be evaluates in future work, in particular on real data sets.
- The main issue is that the paper is only marginally related to missing values, since here the missing values are artifically created to facilitate training. So relevance for this particular workshop is not obvious.

---

### Decision · Program_Chairs · 2020-07-02

**Decision:**

Accept

**Comment:**

We're happy to accept this paper at Artemiss. We'll contact you soon to inform you about more details concerning the format of your presentation at the workshop, and the camera-ready version deadline. Please take into account the referee's comments to write the camera-ready version.